# MUSKETEER: JOINT TRAINING/INFERENCE FOR MULTI-TASK VISION-LANGUAGE MODEL WITH TASK EXPLANATION PROMPTS

## ABSTRACT

We present a sequence-to-sequence vision-language model whose parameters are jointly trained on all tasks and fully shared among multiple tasks, resulting in a single model which we named Musketeer. The integration of knowledge across heterogeneous tasks is enabled by a novel feature called Task Explanation Prompt (TEP). TEP reduces interference among tasks, allowing the model to focus on their shared structure. With a single model, Musketeer achieves results comparable to or better than strong baselines trained on single tasks, almost uniformly across multiple tasks.

## 1 INTRODUCTION

Multi-task training of a homogeneous architecture can be beneficial when tasks are synergistic. Language models such as ChatGPT (OpenAI, 2023) and InstructGPT (Ouyang et al., 2022) benefit from the fact that all tasks in which they are trained share the same data space (input), representation space (architecture), and hypothesis space (output). Their objectives can be seamlessly unified (Paolini et al., 2021) and shared knowledge can be encoded in the weights of a single backbone. In vision, however, tasks can be *heterogeneous* and possibly antagonistic: Different tasks have different hypothesis spaces. Furthermore, multi-modal tasks can have different input spaces. For instance, *Visual grounding* requires mapping images onto semantic classes and their corresponding locations or bounding boxes; *visual question answering* (VQA) maps an image and a string of text representing a question onto a string of text representing the answer. Sharing knowledge among such diverse tasks presents technical challenges, since their hypothesis spaces are not directly comparable. Even when mapped to a shared representation, heterogeneous tasks can interfere with each other, when variability that is useful for a task is detrimental to another. Recent foundation models aiming to support a wide variety of downstream tasks have separate "strong heads" tailored to different tasks, resembling a model zoo built on an embedding with only part of the parameters shared among tasks. This (i) limits the harvesting and exploitation of information shared among tasks, (ii) causes an increase in model complexity, and (iii) limits the extensibility to tasks beyond those for which the individual heads were trained.

We aim to design a jointly-trained vision-language model that can be trained jointly on multiple tasks, based on a representation learned by a common encoder-decoder architecture with fully shared parameters (Fig. 2). The benefit would be shared structure, formats, and information across tasks that (i) improves performance, ideally making the jointly-trained models as good as specialists trained on each single task and (ii) reduces model complexity through parameter sharing.

Towards these goals, we present Musketeer: a jointly-trained vision-language model that can perform multiple tasks without task-specific heads and fine-tuning, while achieving competitive performance compared to previous jointly-trained models (Tab. 4) and even single-task-fine-tuned specialist models (Tab. 2). Achieving the above goals requires developing novel methods to avert task interference. Rather than forcing task separation rigidly through the design of the architecture, we propose to train the model so that it can instantiate task-specific processing pathways at inference time using semantically rich Task Explanation Prompts (TEPs). TEPs are structured text explanations, fed to the model both at training and inference time, that describe input and output spaces, datasets and their format, and instance prompts (Fig. 1). TEP tokens leverage structural semantic information using natural language to guide the training and inference processes. This allows

Figure 1: Example of TEP and baseline prompts for visual grounding. One-hot Prompt: representing task as a fixed vector. Base Prompt: standard prompting adopted by prior arts (Wang et al., 2022; Lu et al., 2022).

Musketeer to avoid task interference not by forcing *task specialization* in the architecture, but by fostering *informational specialization* in the trained model, so task-specific processing pathways inside the trained model can be accessed at inference time by choosing a proper TEP.

## 1.1 KEY CONTRIBUTIONS IN RELATION TO PRIOR WORK

Recent vision-language models at scale can be broadly categorized into four groups: 1) Encoders like CLIP (Radford et al., 2021b) that can be used as the backbone but do not themselves directly address most of the downstream tasks. 2) Systems like Florence (Yuan et al., 2021) that share a core visual encoder but still have separate heavy decoders for individual downstream tasks. 3) Models such as OFA (Wang et al., 2022) with a common architecture that can be jointly pre-trained, but separately fine-tuned on individual tasks without sharing the same encoder-decoder parameters. 4) Frameworks such as Unified-IO (Lu et al., 2022) and UniTAB (Yang et al., 2022) that do have a common backbone with shared parameters, but fall short in achieving competitive performance to single-task-tuned models because of the task interference.

Some of these models, once pre-trained, can be used for downstream tasks, but require task-specific adapters to modify the architecture/parameter for a particular task. This creates a discrepancy between pre-training and fine-tuning, the latter effectively performed on a different architecture. Well-trained adapters tend to be task-specific and not easily transferable across tasks. Even without adapters, task-specific fine-tuning can be expensive, especially if it needs to be performed for multiple downstream tasks, spawning multiple task-specific models, each of which has lost its multi-tasking ability.

Musketeer is architecturally similar to sequence-to-sequence foundation models such as OFA (Wang et al., 2022) and Pixel2Seq (Chen et al., 2021; 2022) that also forgo task-specific adapters in favor of a fully-shared model. However, in our experiments we have found that previous unified VL frameworks (Chen et al., 2021; 2022; Wang et al., 2022) do not achieve high performance in multi-tasking due to the inability of their prompts to manage interference between tasks (See BaseP results in Tab. 3). Therefore, task-specific fine-tuning in OFA (Wang et al., 2022) is still necessary to avoid substantial degradation of downstream task performance. This is especially cogent in multi-modal models that unify tasks as general question-answering problems, where the input question is often insufficient to characterize the task and differentiate it from other tasks on which the model has been trained. For example, in visual grounding of some concept $\mathbb{V}$ , the prompt "Which region does the text $\mathbb{V}$ describe" requires the model to interpret "find" and represent the word "region" with sets of coordinates on the image plane, which do not have a meaningful (topologically consistent) representation in natural language.

If we are to integrate vision-language tasks under the same representation, which in our case is tokenized and processed with a Transformer-based architecture (Vaswani et al., 2017), we need to frame each task in a way that specifies, **as precisely and unambiguously as possible, heterogeneous hypothesis spaces, data formats and configurations, using textual tokens**. This is the

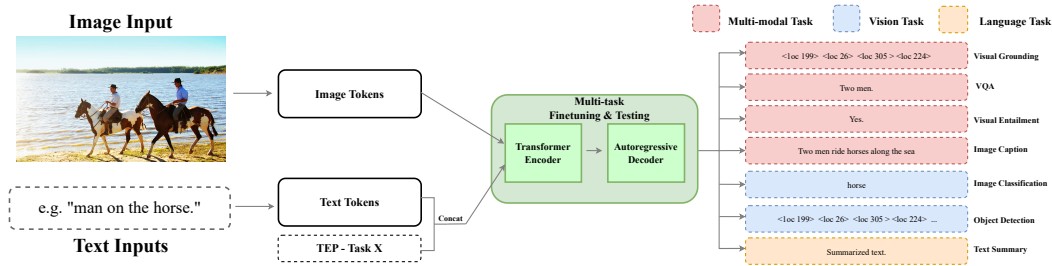

Figure 2: Pipeline overview of multi-tasking of Musketeer. "TEP-Task X" denotes Task Explanation Prompt (TEP) for a specific task, *e.g.*, visual grounding. After Multi-task fine-tuning, Musketeer is capable of performing a variety of tasks under a single architecture and fully-shared parameters in a sequence-to-sequence manner. Each task is specified by a structural Task Explanation Prompt, which provides explicit instructions for conducting each specific task.

key idea behind Task Explanation Prompts: Use the intrinsic structural tasking specification with semantics of natural language to induce the model to align and separate tasks depending on their degree of synergy or interference. It is not just a matter of providing the model with information about which task is to be performed: If this is done with nameless labels, for instance, one-hot vectors corresponding to a selection of discrete configuration, performance is degraded, as we show in our ablation studies (Tab. 6). The rich and structured information in TEPs includes descriptions of the dataset, the input and output format for each task, and an output description of the task target (Fig. 1). The resulting model not only improves efficiency through parameter sharing, but performs on-par or better than each of the specialized models, in all but one evaluations we have performed.

Through the use of TEPs, Musketeer is able to harvest synergistic information in heterogeneous tasks, including visual grounding, image classification, visual entailment, image captioning, visual question answering, and text summarization. In summary, our contributions are

- A jointly trained sequence-to-sequence model with all tasks contributing to training a single shared backbone, where parameters shared across all tasks, is shown in Fig. 2.
- We introduce a novel approach to controlling interference among heterogeneous multi-modal tasks by leveraging structural tasking specifications, using Task Explanation Prompts (Fig. 1). TEPs foster task-specific processing pathways without the need for specialized architectural modules or heads. Fig. 3 provides a motivation of TEP by demonstrating the intrinsic similarity of multi-modal tasks in specific aspects by leveraging sharing knowledge using subprompts.
- An empirical analysis of the proposed model, Musketeer, that illustrates TEP's effectiveness compared to other prompt baselines (Tab. 3, 7 and 10) while retaining high performance even on small data by concurrent training (Tab. 8).

In an ablation analysis (Tab. 6), we also illustrate the critical role of the explanation prompt in specifying and unifying tasks, allowing the trained model to leverage synergies and minimize interference among tasks.

## 1.2 OTHER RELATED WORK IN BROADER CONTEXT

**Multi-modal Pretraining.** Large Transformer models have found fertile ground in modeling language, which is naturally tokenized. They are typically pre-trained as masked autoencoders on large corpora, and then fine-tuned for specific downstream tasks such as named-entity recognition, relation extraction, or question answering (Mikolov et al., 2010; Graves, 2013; Howard & Ruder, 2018; Devlin et al., 2018; Sutskever et al., 2011; Brown et al., 2020; Raffel et al., 2020; Zhong et al., 2022). This paradigm has more recently been adopted in modeling sensory data such as sound or images, despite the absence of a natural tokenization (Chen et al., 2020; Hendricks et al., 2021; Tan & Bansal, 2019; Wang et al., 2021a; Alayrac et al., 2020; Jain et al., 2021; Jia et al., 2021; Radford et al., 2021b). However, unlike in language, these models do not exhibit the same few-shot prowess due to task interference, which prompts the need to fine-tune them, which at the scale of current model is often prohibitively expensive. This has spawned a variety of expedients including task adapters (Yuan et al., 2021; Li et al., 2022b), trained on frozen pre-trained backbones (Tsimpoukelli

et al., 2021; Li et al., 2023). However, such adapters are saturated by task-specific information and decrease, rather than improve, transferability as more task adapters are used.

**Prompt-based Learning.** To enhance the effectiveness of pre-trained models, prompt-oriented fine-tuning has gained traction in the NLP community. This approach reformulates the objective of downstream tasks to be more similar to that of pre-training by inserting manually designed (Schick & Schütze, 2021; 2020), automatically searched (Jiang et al., 2020) or tunable soft prompt tokens (Li & Liang, 2021) with adapters (Hu et al., 2022) into the input text. One recent work, ProQA (Zhong et al., 2022), utilizes a structural prompt to distinguish different Question Answering (QA) tasks. While Musketeer also employs TEP in a structural form, we differ by exploring its effectiveness in a wider range of multi-modal and multi-tasking scenarios.

Prompt-tuning has demonstrated that large language models can learn effectively in context and improve performance on few-shot tasks. This has motivated the development of prompt tuning methods for multi-modal pretrained models, which incorporate modifications or adapters for prompts into either frozen language models (Tsimpoukelli et al., 2021) or CLIP (Radford et al., 2021a)-like models (Gao et al., 2021; Zhang et al., 2021). These methods aim to adapt the pre-trained model to the specific task more efficiently, in contrast to Musketeer which focuses on a joint model without task-specific adaptation.

**Unified Frameworks.** Task-specific adapters can be used to train a model on multiple tasks, but task-specific information is not shared in the core model. To facilitate transfer across tasks, (Kaiser et al., 2017) propose a uniform format for representing tasks. Others unify heterogenous tasks in different modalities: VLT-5 (Cho et al., 2021) and UNICORN (Yang et al., 2021) have demonstrated text-generation-based multi-modal pretrained models. Meanwhile, PERCEIVER (Jaegle et al., 2021) and PERCEIVERIO (Jaegle et al.) propose a simple framework that can process information from multiple modalities with a uniform byte-sequence representation. Other approaches, such as UNIT (Hu & Singh, 2021) and FLAVA (Singh et al., 2021), unify tasks across different modalities by designing various task-specific layers. Cross-task fine-tuning has also been shown effective in model-based reinforcement learning (Xu et al., 2023). All these approaches require task-specific adapters and fine-tuning tailored to the specific task at hand.

Other works have aimed to derive a joint model that can handle multiple tasks without single task fine-tuning. For instance, (Aghajanyan et al., 2021; Wei et al., 2021; Aribandi et al., 2021) is effective at multi-task fine-tuning on language tasks, but unable to handle vision tasks which are more diverse in format. Flamingo (Alayrac et al., 2022) takes visual inputs, but only supports text generation as outputs, while Uni-Perceiver (Zhu et al., 2021; Li et al., 2022a) is a contrastive model that does not support text-to-text generation. In contrast, Musketeer achieve multitasking without task-specific architectures or task-specific fine-tuning.

## 2 MUSKETEER

### 2.1 TASKS & DATASETS

**Diverse Tasks.** Musketeer is trained on seven distinct tasks, each embodied in a different dataset and corresponding benchmark:

- Image Caption (COCO (Chen et al., 2015)): The input is a single image, and the output is a string of text that accurately describes it, as assessed by human annotators.
- Visual Grounding (RefCOCO (Yu et al., 2016; Mao et al., 2016)): The input is an image and a text query, and the output is the location on the image that corresponds to the query in the form of the coordinates of a bounding box.
- Visual Entailment (SNLI-VE (Xie et al., 2019)): The input is an image and a textual premise about the input image, and the output is a ternary decision on whether the hypothesis is supported by the image (yes / no / maybe).
- Visual Question Answering (VQA) (VQAv2 (Goyal et al., 2017)): The input is an image and a question in textual form, and the output is a textual answer to the question.
- Image Classification (ImageNet (Deng et al., 2009)), where the model is required to assign an input image to one of a predefined set of classes.

- Text Summarization (Gigaword (Rush et al., 2015)): The input is a text, and the output is the abstractive summarization of the given input text.
- Object Detection (COCO (Chen et al., 2015)): The input is an image, and the output is a textual representation of the coordinates of a bounding box, along with a class label for the object contained within, assumed to be from a finite set of known labels. Following OFA (Wang et al., 2022), we use the detection task only for training.

We train Musketeer on all these tasks jointly, and evaluated the model performance on each task, using the benchmark corresponding to the datasets quoted, in comparison with models specifically adapted or fine-tuned for these tasks.

## 2.2 TASK EXPLANATION PROMPT

For a jointly-trained model to be capable of performing different tasks without needing task-specific adapters or fine-tuning, it needs to have enough task-specific data to fully understand and differentiate each task, as well as to determine which task to conduct. To address this issue, prior **NLP** arts (McCann et al., 2018; Raffel et al., 2020) have adopted a prompting paradigm by adding a task description to input sequences, such as asking "*What is the summary?*" However, this can be challenging in the context of unifying **multi-modal** tasks that involve signals such as images, which require inference of physical properties. For instance, recent multi-modal jointly-trained methods (Yang et al., 2022; Lu et al., 2022) have shown apparent performance drops in multi-tasking training compared to models fine-tuned for specific tasks, despite the use of standard prompting schemes. To tackle above issues, we propose a novel approach that avoids rigid task separation in the model's architecture. Instead, we suggest training the model to instantiate task-specific processing pathways by utilizing semantically rich Task Explanation Prompts (TEP). As illustrated in Fig. 1, the TEP is a structural prompt consisting of five key components covering detailed instruc-

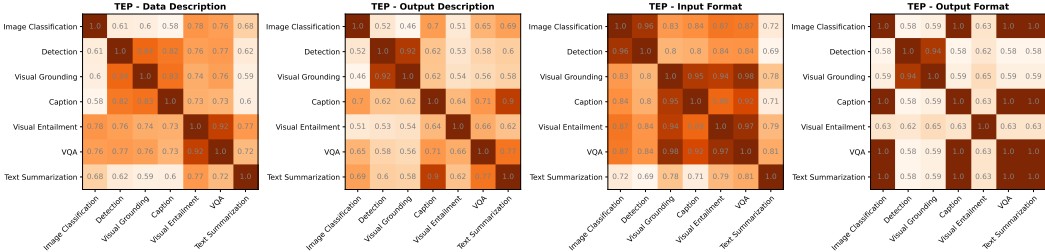

Figure 3: TEP subprompts' similarity matrices. They are constructed by computing cosine distances between TEP subprompts, which are obtained by inputting TEP subprompts into a language model. These matrices demonstrate the similarities among TEP subprompts across various tasks. The similarity matrices of different types of prompts are included in Supplementary.

- **Data Description:** Description of how this dataset is built, including information of data source, dataset contents and labeling specification, which is generated by ChatGPT and verified by human.
- **Input Format:** Summary of how the input multi-modal sequence is formulated. (*E.g.*, by concatenating prompts and image features for image caption, or prompts, text and image features for visual grounding.)
- **Output Format:** Specification of how the output sequence is expected to formulate. (*E.g.*, a word in finite set for image classification, or four coordinate tags describing a region for visual grounding.)
- **Output Description:** Detailed description of generation targets to specify Output Format.
- **Instance Prompt:** Conclusive short prompt containing the input text (if any).

As explained above, TEP uses structural natural language descriptions to define the dataset domain (Data Description), specify the input/output format of modalities used in a particular task, clarify the desired output (Output Description), and provide an overview of the current task (Instance Prompt). These guidelines reduce interference among tasks by specifying differences and similarities of tasks in data and hypothesis space. For example, as illustrated in Fig. 3, TEP specifies the differences between the output spaces of tasks such as text summarization and image classification, ensuring

Table 1: Tasks and corresponding datasets used to validate the feasibility of multi-task fine-tuning. "566k (120k)" denotes expanded dataset from 120k samples to 566k by random tiling. "ratio" refers to the proportion of the original training set that is utilized for training data.

| Task | Image Classification | | Object Detection | | Image Caption | | Visual Entailment | | Visual Grounding | | VQA | | Text Summarization | | Total |
|---|---|---|---|---|---|---|---|---|---|---|---|---|---|---|---|
| Dataset | ImageNet | ratio | COCO | ratio | COCO | ratio | SNLI-VE | ratio | RefCOCO | ratio | VQAv2 | ratio | Gigaword | ratio | - |
| Subset$_{small}$ | 50k | 4% | 50k | 42% | 50k | 9% | 50k | 9% | 50k | 41% | 50k | 4% | 50k | 1% | 350k |
| Subset$_{vg}$ | 121k | 10% | 121k (118k) | 100% | 121k | 21% | 121k | 22% | 121k | 100% | 121k | 9% | 121k | 3% | 847k |
| Subset$_{caption}$ | 566k | 47% | 566k (118k) | 100% | 566k | 100% | 566k (529k) | 100% | 566k (121k) | 100% | 566k | 44% | 566k | 15% | 3962k |

that they do not interfere with each other. On the other hand, TEP enables sharing of commonly required information among tasks. For instance, if multiple tasks are required to output localization bounding boxes (*e.g.*, Visual Grounding and Detection), output formats of TEP for these tasks would be similar, allowing sharing information among them. TEP outperforms other candidate subprompts like soft prompts (learnable vectors (Li & Liang, 2021; Zhong et al., 2022)), and is compared to two baseline prompts in Tab. 3 . More discussions on the selection of TEP subprompts can be found in Section 3.5.

## 2.3 ARCHITECTURE & TRAINING

**Unified Architecture.** We use an encoder-decoder architecture without task-specific heads as backbone to our model (as shown in Fig. 2). Our encoder and decoder consist of stacked Transformer layers, each composed of self-attention Transformers, cross-attention Transformers (only in decoders), and feed-forward neural networks (FFN). We use residual connections and head scaling for self-attention to stabilize training. Layer Normalization is applied before each multi-head self-attention layer and FFN. To maintain the positional information of words, we employ separate parameterizations for computing word contextual correlation and positional correlation, and then combine them additively, as described in (Ke et al., 2020). We also follow the practice of previous models (Radford et al., 2018; Lewis et al., 2020) by applying byte-pair encoding (BPE) (Sennrich et al., 2016) to the given text sequence to convert it into a subword sequence, which we then embed into features. For image data, we use a pretrained ResNet (He et al., 2016) to convert the input image into a feature of the hidden layer size, following the approach of (Dai et al., 2021; Wang et al., 2021b; 2022). In the encoding process, the preprocessed text, image, and Task Explanation Prompt tokens will be concatenated together in a sequential manner to form a single input sequence for the encoder. Following (Chen et al., 2022; Wang et al., 2022), information from different modalities is shared within a global multi-modal vocabulary across all tasks, and no parametric task-specific adapters will be added to downstream tasks.

**Balanced Sampling.** To avoid the data imbalance and overfitting to any particular task, we adopt a balanced sampling strategy during training. Specifically, we sample equal number of data from all seven tasks at each iteration, and then conduct individual forward propagation for each task in a single mini-batch, enabling their corresponding pre- and post-processing steps (such as Trie-based search (Cormen et al., 2009; Wang et al., 2022) for classification tasks). The gradients computed for all tasks are then aggregated for the final update, as suggested in (Aghajanyan et al., 2021), to ensure that every task is involved in the gradient updating process.

**Joint Optimization.** Instead of using task-specific loss functions, which can improve performance on individual tasks, we opted for a standard cross-entropy loss that is applied to all tasks. This decision simplifies the training process and creates a more homogeneous space for losses across tasks. This allows for a more direct comparison of task performance and avoids the issue of specific tasks dominating the gradients. We also choose to forego multi-stage fine-tuning (like (Aghajanyan et al., 2021; Yang et al., 2022)) and task-specific hyperparameters tuning, which may potentially achieve better performance but require considerable human effort.

**Inference.** For tasks evaluated here, we perform auto-regressive sequence prediction. Additionally, for classification, visual entailment, and VQA tasks, we adopt Trie (Cormen et al., 2009) post-processing, as recommended in OFA (Wang et al., 2022), to avoid generating invalid label outside the closed set.

## 3 EXPERIMENTS

In this section, we first outline the composition of the training dataset for various scales and the training method for Musketeer. Next, we present the multi-tasking capability of Musketeer and its comparative results against the specialist OFA model and other advanced multi-tasking models that

Table 2: Performance comparison to OFA specialist models reported in (Wang et al., 2022), which are noted as "OFA-VG" (for visual grounding), "OFA-VE" (for visual entailment), and "OFA-Cap" (for caption). Musketeer support multi-tasking and have been shown to achieve comparable or even superior performance to specialist models which can only perform a specific task. Please note that we only perform single stage joint training, without extra CIDEr optimization stage which may further improve caption performance. "data usage" refers to the proportion of the original training set that is utilized for training data.

| Model | Training Set | Multi-Tasking | Visual Grounding | | | | Visual Entailment | | | Caption | | |
|---|---|---|---|---|---|---|---|---|---|---|---|---|
| | | | data usage | val | test-A | test-B | data usage | dev | test | data usage | B@4 | CIDEr |
| OFA$_{Base}$-VG | RefCOCO | ✗ | 100% | 88.5 | 90.7 | 83.3 | - | - | - | - | - | - |
| OFA$_{Base}$-VE | SNLI-VE | ✗ | - | - | - | - | 100% | **89.3** | **89.2** | - | - | - |
| OFA$_{Base}$-Cap | COCO (caption) | ✗ | - | - | - | - | - | - | - | 100% | **41.0** | **138.2** |
| Musketeer$_{Base}$ | Subset$_{vg}$ | ✓ | 100% | 88.4 | 91.0 | 84.4 | 22% | 86.3 | 86.7 | 21% | 38.7 | 130.7 |
| Musketeer$_{Base}$ | Subset$_{caption}$ | ✓ | 100% | **88.7** | **91.2** | **85.5** | 100% | 89.2 | 89.1 | 100% | 40.9 | 137.2 |
| OFA$_{Large}$-VG | RefCOCO | ✗ | 100% | 90.1 | 92.9 | 85.3 | - | - | - | - | - | - |
| OFA$_{Large}$-VE | SNLI-VE | ✗ | - | - | - | - | 100% | **90.3** | 90.2 | - | - | - |
| OFA$_{Large}$-Cap | COCO (caption) | ✗ | - | - | - | - | - | - | - | 100% | 42.4 | **142.2** |
| Musketeer$_{Large}$ | Subset$_{vg}$ | ✓ | 100% | 90.7 | 93.2 | 86.9 | 22% | 88.7 | 88.5 | 21% | 40.7 | 136.9 |
| Musketeer$_{Large}$ | Subset$_{caption}$ | ✓ | 100% | **90.8** | **93.1** | **87.6** | 100% | 89.9 | **90.2** | 100% | **42.5** | 140.2 |

Table 3: Evaluation of multi-tasking performance of Musketeer. TEP outperforms other prompts consistently, making TEP demonstrates competitive performance across tasks, despite no task-specific fine-tuning or adaptation. B@4, R-1, R-2 and R-L denote BLEU@4, ROUGE-1, ROUGE-2, and ROUGE-L respectively.

| Model | Training Set | Prompt Type | Visual Grounding | | | Visual Entailment | | VQA | Caption | | Image Classification | Text Summary | | |
|---|---|---|---|---|---|---|---|---|---|---|---|---|---|---|
| | | | val | test-A | test-B | dev | test | test-dev | B@4 | CIDEr | | R-1 | R-2 | R-L |
| Musketeer$_{Base}$ | Subset$_{small}$ | one-hot | 84.2 | 87.1 | 80.3 | 82.6 | 82.1 | 68.5 | 35.9 | 120.1 | 52.2 | 32.7 | 14.2 | 30.4 |
| | | BaseP | 85.8 | 88.8 | 81.7 | 83.4 | 83.5 | 69.2 | 37.3 | 125.5 | 53.4 | 33.1 | 14.7 | 30.8 |
| | | TEP | **87.5** | **90.3** | **83.1** | **84.9** | **84.5** | **70.6** | **38.3** | **130.3** | **56.5** | **33.4** | **15.1** | **31.1** |
| Musketeer$_{Base}$ | Subset$_{vg}$ | one-hot | 86.1 | 87.8 | 81.2 | 84.2 | 84 | 68.8 | 36.5 | 123.7 | 58.2 | 33.9 | 15.2 | 31.4 |
| | | BaseP | 87.5 | 90.1 | 83.4 | 85.1 | 85.0 | 69.6 | 37.6 | 127.8 | 59.1 | 34.2 | 15.6 | 31.8 |
| | | TEP | **88.4** | **91.0** | **84.4** | **86.3** | **86.7** | **71.4** | **38.7** | **130.7** | **62.1** | **34.5** | **16.0** | **32.3** |
| Musketeer$_{Base}$ | Subset$_{caption}$ | one-hot | 86.2 | 87.7 | 82.2 | 85.9 | 85.5 | 69.9 | 37.6 | 128.8 | 59.7 | 34.4 | 16.0 | 32.2 |
| | | BaseP | 87.6 | 90.4 | 83.3 | 87.2 | 86.9 | 70.4 | 38.8 | 134.2 | 60.4 | 34.9 | 16.4 | 32.4 |
| | | TEP | **88.7** | **91.2** | **85.5** | **89.2** | **89.1** | **72.0** | **40.9** | **137.2** | **62.9** | **35.1** | **16.7** | **32.8** |
| Musketeer$_{Large}$ | Subset$_{small}$ | BaseP | 89 | 92 | 84.3 | 85.9 | 86.0 | 73.2 | 38.3 | 130.4 | 63.2 | 34.5 | 15.9 | 32.1 |
| | | TEP | **89.7** | **92.3** | **86.0** | **87.5** | **87.2** | **74.1** | **40.3** | **135.7** | **65.6** | **34.8** | **16.2** | **32.3** |
| Musketeer$_{Large}$ | Subset$_{vg}$ | BaseP | 90.1 | 92.4 | 85.9 | 87.8 | 87.7 | 73.7 | 39.5 | 133.2 | 67.4 | 35.2 | 16.4 | 32.6 |
| | | TEP | **90.7** | **93.2** | **86.9** | **88.7** | **88.5** | **74.7** | **40.7** | **136.9** | **69.7** | **35.4** | **16.9** | **33.1** |
| Musketeer$_{Large}$ | Subset$_{caption}$ | BaseP | 90.2 | 92.6 | 86.0 | 88.0 | 87.9 | 74.1 | 40.9 | 137.9 | 68.1 | 35.7 | 16.9 | 33.2 |
| | | TEP | **90.8** | **93.1** | **87.6** | **89.9** | **90.2** | **75.0** | **42.5** | **140.2** | **70.2** | **36.0** | **17.3** | **33.5** |

do not require task-specific fine-tuning. Additionally, we show the performance of Musketeer can be improved by including more tasks in the training set. Finally we conduct ablation studies to analyze the functioning mechanism of TEP.

## 3.1 TRAINING DATASET COMPOSITION

Given the significant variation in training dataset size across seven tasks (ranging from 118k for SNLI-VE to 1.3M for VQAv2), jointly training Musketeer with all data results in data imbalance, leading to inadequate multi-tasking ability. To address this, we sample equivalent amounts of training data for each task. We also create subsets in varying sizes, as outlined in Tab. 1, we train Musketeer on subsets in three scales in terms of total number of samples, **Subset$_{small}$**: consists of 50k samples for each task. **Subset$_{vg}$**: consists of 120k samples for each task and contains the entire RefCOCO dataset for visual grounding. **Subset$_{caption}$**: consists of 560k samples for each task and contains the entire COCO dataset for image captioning.

## 3.2 EXPERIMENTAL SETUP

Unlike (Wang et al., 2022), we directly evaluate the joint-task trained model without any task-specific fine-tuning. As suggested in (Yang et al., 2022; Wang et al., 2022; Li et al., 2022a), we initialize weights of Musketeer from pretrained model in (Wang et al., 2022). During joint training, all images are cropped to a size of $480 \times 480$, with $16 \times 16$ patches. The maximum text sequence length of both the encoder and decoder is set to 512 as in (Wang et al., 2022). Our optimizer of choice is AdamW (Loshchilov & Hutter, 2019), with $(\beta 1, \beta 2, \epsilon) = (0.9, 0.999, 1e-8)$ and a learning rate of $1e-4$, which is controlled by a linearly decayed scheduler with a warmup ratio of 0.01. We further apply dropout regularization with a ratio of 0.1 and weight decay of 0.01 during training. For more information on our implementation, please refer to the Supplementary.

## 3.3 EFFECTIVENESS OF MUSKETEER

In this section, we firstly compare our proposed Musketeer with OFA specialist models. Then our investigation focuses on the effectiveness of the Task Explanation Prompt (TEP) in comparison to baseline prompts across 6 diverse tasks with various scales of training data and model sizes.

Table 4: Comparison of Musketeer with other jointly-trained models without task-specific fine-tuning. #Param is the number of trainable model parameters. Musketeer outperforms the methods listed uniformly, despite its relatively compact size. Please note that we only report the results without any task-specific fine-tuning (consistent to Musketeer) and both Musketeer models are trained on the Subset$_{caption}$ dataset, which means Musketeer only utilizes 44% of VQAv2 training data.

| Model | #Param | Visual Grounding | | | Visual Entailment | | Caption | | VQA |
|---|---|---|---|---|---|---|---|---|---|
| | | val | test-A | test-B | dev | test | B@4 | CIDEr | |
| Pixel2seq-v2 (Chen et al., 2022) | 132M | - | - | - | - | - | 34.9 | - | - |
| Flamingo (Alayrac et al., 2022) | - | - | - | - | - | - | - | 113.8 | - |
| UniTAB (Yang et al., 2022) | - | 88.5 | - | - | - | - | - | 115.8 | 69.1 |
| Unified-IO$_{Base}$ (Lu et al., 2022) | 241M | - | - | - | 85.6 | - | - | - | 61.8 |
| Unified-IO$_{Large}$ (Lu et al., 2022) | 776M | - | - | - | 86.1 | - | - | - | 67.8 |
| Unified-IO$_{XLarge}$ (Lu et al., 2022) | 2,925M | - | - | - | **91.1** | - | - | 126.3 | **77.9** |
| Uni-Perceiver-v2$_{Base}$ (Li et al., 2022a) | 308M | - | - | - | - | - | 35.4 | 116.9 | - |
| Uni-Perceiver-v2$_{Large}$ (Li et al., 2022a) | 446M | - | - | - | - | - | 36.5 | 122.5 | - |
| Musketeer$_{Base}$ | 182M | 88.7 | 91.2 | 85.5 | 89.2 | 89.1 | 40.9 | 137.2 | 72.0 |
| Musketeer$_{Large}$ | 472M | **90.8** | **93.1** | **87.6** | 89.9 | **90.2** | **42.5** | **140.2** | 75.0 |

Table 5: Ablation on Musketeer trained with varying task numbers on Subset$_{small}$ (50k samples for each task) . "#Task=1" denote specialist models which is evaluated on their corresponding training tasks (3 models in total). "#Task=3,5,7" denotes multi-task fine-tuned models. For the tasks we use here, please refer to Supplementary.

| #Task | Visual Grounding | | | Visual Entailment | | Caption | |
|---|---|---|---|---|---|---|---|
| | val | test-A | test-B | dev | test | B@4 | CIDEr |
| 1 (VG) | 86.4 | 88.4 | 81.9 | - | - | - | - |
| 1 (VE) | - | - | - | 84.5 | 84.2 | - | - |
| 1 (Caption) | - | - | - | - | - | 38.2 | 128.9 |
| 3 | 86.0 | 89.0 | 81.3 | 84.6 | 84.5 | 37.1 | 123.9 |
| 5 | 86.3 | 88.9 | 81.8 | 84.5 | **84.6** | 38.2 | 128.5 |
| 7 | **87.5** | **90.3** | **83.1** | **84.9** | 84.5 | **38.3** | **130.3** |

**One for all: Musketeer vs OFA specialist models.** Musketeer uses the same architecture and pretrained weights as OFA (Wang et al., 2022), which is currently the state-of-the-art method for many visual language tasks such as visual grounding (RefCOCO), image captioning (COCO), and visual entailment (SNLI-VE). We show that Musketeer achieves highly competitive multi-tasking performance by comparing its jointly trained model with OFA specialist models in Tab. 2. Unlike OFA specialist models, Musketeer can perform multiple tasks simultaneously without any task-specific fine-tuning. Moreover, our results show that Musketeer could achieve comparable, or even better performance than OFA specialist models. For instance, on the visual grounding task, Musketeer outperforms OFA specialist models across all test splits. Considering OFA is a strong specialist VL baseline model , those findings indicate that Musketeer demonstrates significant efficacy in transferring knowledge across various tasks and attaining superior performance in multitasking.

**Comparison with baseline prompt and one-hot prompt.** To show the effectiveness of TEP in Musketeer, we utilize OFA (Wang et al., 2022) prompt as our baseline prompt (noted as BaseP), which describes the task in one single, straightforward sentence. One could consider Base Prompt trained Musketeer as a natural extension of OFA (Wang et al., 2022) that can perform multiple tasks. Another simple prompt we adopt is one-hot prompt which employs a one-hot vector as the prompt (noted as one-hot). Results in Tab. 3 show that TEP consistently outperforms the other prompts, regardless of task type, model size, or training dataset scale.

## 3.4 COMPARISON WITH STATE-OF-THE-ART METHODS

We present the performance results of Musketeer, along with other multi-task models, in Tab. 4. Our exclusive attention is directed towards multi-task performance, and we present the results for all models without any task-specific fine-tuning. Musketeer is trained on Subset$_{caption}$, which contains 100% of the data for visual grounding, visual entailment, and image captioning. It shows that Musketeer surpasses other multi-task models substantially across several tasks, affirming the effectiveness of it. The only exception is that Unified-IO$_{XLarge}$ performs better than Musketeer on visual entailment and VQA. Nonetheless, it's worth noting that Unified-IO$_{XLarge}$ has a significantly larger (6.2 times) model size than Musketeer. Besides, it uses 100% of the VQA training data, while Musketeer only utilizes 44%.

Table 6: Ablations on specific TEP subprompts. We report performance of Musketeer$_{Base}$ trained on Subset$_{small}$ with varying TEP settings. Best results are achieved by TEP with all four subprompts, suggesting each subprompt's positive contribution to the overall performance.

| Prompt Type | Data Description | I/O Format | Output Description | Instance Prompt | Visual Grounding | | | Visual Entailment | | Caption | |
|---|---|---|---|---|---|---|---|---|---|---|---|
| | | | | | val | test-A | test-B | dev | test | B@4 | CIDEr |
| one-hot | ✗ | ✗ | ✗ | ✗ | 84.2 | 87.1 | 80.3 | 82.6 | 82.1 | 35.9 | 120.1 |
| BaseP | ✗ | ✗ | ✗ | ✓ | 85.8 | 88.8 | 81.7 | 83.4 | 83.5 | 37.3 | 125.5 |
| TEP *w/o* Instance Prompt | ✓ | ✓ | ✓ | ✗ | 85.6 | 88.7 | 81.5 | 83.7 | 83.5 | 36.3 | 126.7 |
| TEP *w/o* I/O | ✓ | ✗ | ✗ | ✓ | 86.7 | 89.5 | 82.7 | 84.3 | 84.2 | 38.2 | 130.0 |
| TEP *w/o* Data Description | ✗ | ✓ | ✓ | ✓ | 87.2 | 90.1 | 83.0 | 84.4 | 84.2 | **38.4** | **130.3** |
| TEP | ✓ | ✓ | ✓ | ✓ | **87.5** | **90.3** | **83.1** | **84.9** | **84.5** | 38.3 | **130.3** |

Table 7: Ablations on other TEP subprompt candidates. None of the 3 listed candidates demonstrate significant performance improvements on all three tasks, therefore not adopted.

| Prompt Type | Visual Grounding | | | Visual Entailment | | Caption | |
|---|---|---|---|---|---|---|---|
| | val | test-A | test-B | dev | test | B@4 | CIDEr |
| BaseP | **85.8** | **88.8** | 81.7 | 83.4 | 83.5 | 37.3 | **125.5** |
| + fixed task vector | 85.1 | 88.2 | 81.0 | 83.5 | 83.6 | 37.1 | 124.9 |
| + learnable task vector | 84.9 | 87.8 | 80.7 | 83.7 | 83.6 | **37.4** | 125.5 |
| + task description (Wiki) | 84.5 | 87.9 | 79.3 | 82.5 | 82.4 | 36.9 | 125.2 |
| + task description (ChatGPT) | 85.7 | 88.5 | **81.9** | **84.1** | **83.9** | 37.0 | 124.5 |
| TEP | **87.5** | **90.3** | **83.1** | 84.9 | 84.5 | **38.3** | **130.3** |
| + fixed task vector | 86.9 | 89.9 | 81.7 | 84.5 | 84.6 | 37.9 | 127.5 |
| + learnable task vector | 86.2 | 88.7 | 80.9 | 83.7 | 83.9 | 37.5 | 125.9 |
| + task description (Wiki) | 87.2 | **90.3** | 83.0 | 84.3 | 84.2 | 38.0 | 128.0 |
| + task description (ChatGPT) | 87.4 | 90.2 | 82.9 | **85.0** | **84.6** | 38.2 | 130.0 |

## 3.5 Ablation Studies

**Joint training/inference under a single model: More tasks, better accuracy.** For multimodal tasks, generally, if the training sample amount for existing tasks remains unchanged, adding new tasks into a multi-modal jointly-trained model training may lead to decreased performance for existing tasks (Lu et al., 2022; Yang et al., 2022). However, when sufficient types of tasks are available for joint training, Musketeer achieves comparable performance to specialist models on existing tasks. Results in Tab. 5 illustrate that for Musketeer, even if the training sample amount for existing tasks remains the same, the addition of more tasks (from 3 tasks to 7 tasks) can still improve the performance of existing tasks. Also, when the task amount increases to 7, multi-task Musketeer can surpass the single-task tuned Musketeer, which is typically considered as the upper-bound performance in previous studies (Lu et al., 2022; Yang et al., 2022).

**Different TEP subprompts.** To assess the effectiveness and significance of each TEP's subprompt (Data Description, I/O Format, Output Description, and Instance Prompt), we conduct experiments on Musketeer by selectively removing TEP subprompts. As shown in Tab. 6, all TEP subprompts are essential to the full TEP performance, while Instance Prompt and I/O are significantly more important than Data Description. Additionally, if the Instance Prompt is removed from TEP (TEP *w/o* Instance Prompt in Tab. 6), it still perform significantly better than one-hot Prompt and produced comparable results to baseline Prompt (BaseP). This implies that, despite a lack of explicit instructions on input text usage , the data and I/O description can still furnish the model with rich information on various tasks.

**Other candidate TEP subprompts.** In addition to these four subprompts, we also experiment three other candidates for TEP, including learnable task vectors (also known as soft prompts (Zhong et al., 2022)), fixed task vectors, and task descriptions (similar to dataset descriptions generated by ChatGPT and verified by humans). Results in Tab. 7 show that incorporating fixed or learnable task vectors results in decreases of the model's performance. Besides, Our experiments do not reveal any substantial performance improvement upon the inclusion of task descriptions. In summary, an effective subprompt for TEP must furnish the model with rich and structured description for the task.

## 4 Conclusion and Discussion

We present a jointly trained vision-language model with all-shared model parameters, Musketeer. Musketeer utilizes longer prompts as input, which may result in around 10%-15% extra latency compared to specialist OFA (see Tab. 14 in Appendix), but multiple specialist models are not required anymore due to using fully-shared model parameters. Additionally, we observed that OFA-based models tend to have low baseline performance on the detection task. Although Musketeers with TEP still outperform specialist OFA, we follow OFA (Wang et al., 2022) paper and choose not to present object detection task as our main results. For more details and discussion, please refer to the supplementary materials.

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

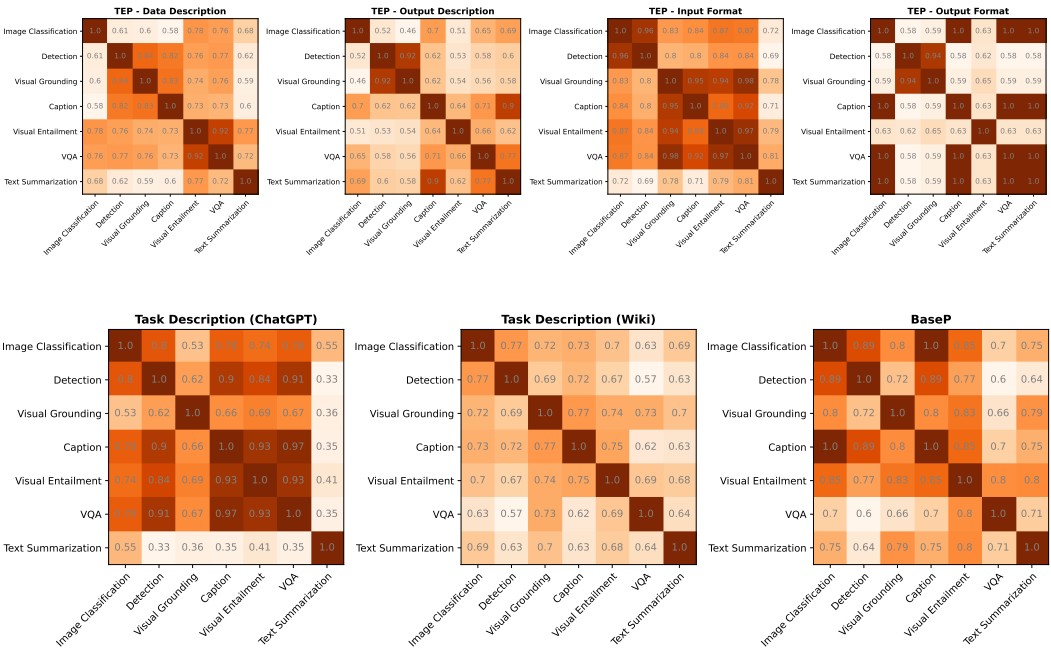

Figure 4: Similarity matrices of subprompts pf TEP and other candidate prompts across seven tasks. They are constructed by computing cosine distances between corresponding prompts, which are obtained by feeding them into a language model.

# A   APPENDIX

## A.1   SIMILARITY MATRICES OF OTHER PROMPTS

We further compare similarity matrices of TEP and other prompts, including Task Description (Wiki), BaseP, and Task Description (ChatGPT (OpenAI, 2023)). As illustrated by Fig. 4, TEP's subprompts are effective at distinguishing different tasks by specifying their differences and shared structure in the data and hypothesis spaces, thereby reducing task interference. In contrast, Task Description (Wiki) is relatively poor at capturing cross-task relationships, leading to a performance drop in multi-tasking scenarios (as shown in Tab. 7 of the main text). Although BaseP and Task Description (ChatGPT) are more informative, they still lack some important task relationships. For example, BaseP fails to see the similarity between VQA and visual entailment, while Task Description (ChatGPT) doesn't capture the relationship between object detection and visual grounding. Our experimental results (Tab. 3 and 7 of the main text) are consistent with these observations, showing that Task Description (Wiki) is not beneficial to multi-tasking performance, while BaseP and Task Description (ChatGPT) are better but still significantly outperformed by TEP.

## A.2   EXTENSIONAL EXPERIMENTS

**Concurrent multi-task finetuning on small data: visual grounding.**   One further question for Musketeer is whether it can perform well on a task with limited training data by incorporating data from other tasks. To explore this question, we trained Musketeer$_{Base}$ on visual grounding with only 32 or 100 samples, while jointly finetuning the model with six other tasks that has 50k samples each. To verify whether Musketeer can leverage cross-task information to improve the performance of task with limited samples, we also exclusively trained Musketeer$_{Base}$ on visual grounding with only 32 or 100 samples for comparison. Our results, as shown in Tab. 8, indicate that multi-task tuned models, including both Base Prompt and TEP prompted ones, exhibit significantly better performance than non-multi-task tuned models in such scenario. Such results indicate that Musketeer can enhance performance on a task with limited samples by utilizing knowledge from multiple tasks. Moreover, the TEP-prompted model outperforms Base Prompted (BaseP) models by around 3% in case of

using 32 VG samples and 4% for 100 VG samples case, which indicates that TEP is a more potent prompt for small data with concurrent training configuration.

Table 8: Experiments on concurrent multi-task training for small data. "Multitask tuning" denotes the model is jointly trained with other 6 tasks in Subset$_{small}$. Multi-task tuned models, including both Base- and TEP-prompted models, exhibit superior performance over specialist models in this scenario.

| Prompt Type | Multitask tuning | # VG Samples | val | test-A | test-B |
|---|---|---|---|---|---|
| TEP | ✓ | 50k | 87.5 | 90.3 | 83.1 |
| BaseP | ✗ | 32 | 18.2 | 23.3 | 13.4 |
| BaseP | ✓ | 32 | 69.1 | 75.2 | 61.9 |
| TEP | ✓ | 32 | **72.1** | **79.3** | **65.8** |
| BaseP | ✗ | 100 | 20.5 | 25.3 | 15.3 |
| BaseP | ✓ | 100 | 75.1 | 80.3 | 70.5 |
| TEP | ✓ | 100 | **79.2** | **84.5** | **74.1** |

**Concurrent multi-task finetuning on small data: visual entailment.** This section provides extra experiments on visual entailment for Musketeer to see whether it can perform well on a task with limited training data by incorporating data from other tasks. In addition to Sec. 3.6 in the main text, we trained Musketeer$_{Base}$ on visual entailment with only 32 or 100 samples, while jointly finetuning the model with six other tasks that has 50k samples each. Our results, as shown in Tab. 9, consistent to former experiments and further indicate that TEP-prompted Musketeer can better enhance performance on a task with limited samples by utilizing knowledge from multiple tasks.

Table 9: Experiments on concurrent multi-task training for small data of visual entailment. "Multitask tuning" denotes the model is jointly trained with other 6 tasks in Subset$_{small}$. TEP-prompted models exhibit superior performance over Base-prompted models in this scenario.

| Prompt Type | Multitask tuning | # VE Samples | val | test |
|---|---|---|---|---|
| TEP | ✓ | 50k | 84.9 | 84.5 |
| BaseP | ✓ | 32 | 67.2 | 67.8 |
| TEP | ✓ | 32 | **69.4** | **69.5** |
| BaseP | ✓ | 100 | 72.6 | 72.4 |
| TEP | ✓ | 100 | **73.5** | **73.8** |

**Replacing TEP subprompts with one-hot vector.** To verify the importance of structured text explanations in TEP, we replace subprompts except Instance Prompt in TEP with one-hot vectors, which is noted as TEP-one-hot in Tab. 10. TEP-one-hot is still a structured prompt but removes the detailed textual description. It's worth noting that tasks with homogeneous I/O formats share the same one-hot vector (e.g., VQA and image caption). Results in Tab. 10 shows that using TEP-one-hot leads to a decrease in performance compared with TEP. These results indicate that more text explanations with structure can boost the model performance. However, we find that TEP-one-hot outperforms BaseP, which demonstrates that structured prompts has superiority to prompts that lack structured information.

**TEP's effectiveness on other architecture** We utilized the official released VL-T5 code to train a 3-Task model encompassing Visual Question Answering (VQA), Visual Grounding, and Image Captioning tasks, incorporating the TEP. The resulting model, called VL-T5-All-TEP, was compared against VL-T5 models trained on individual tasks as well as all tasks combined. Tab. 11 demonstrates the performance of the models. In the case of Visual Grounding (VG) and VQA tasks, the TEP-trained VL-T5 model surpasses the results achieved by the models trained on single tasks. For the Captioning task, TEP outperforms VL-T5-All (which uses a simple one-word prompt) and achieves comparable results to those obtained by the single task model.

**Results on object detection.** Tab. 12 presents the object detection results on the COCO dataset for both the single-task-tuned OFA and multitask-tuned Musketeers. As previously reported, the TEP model consistently outperforms the BaseP model, demonstrating competitive performance across tasks even without task-specific fine-tuning. However, we noticed that OFA-based models tend to

Table 10: Ablations on replacing all TEP subprompts except Instance Prompt with one-hot vector.

| Prompt Type | Visual Grounding | | | Visual Entailment | | Caption | |
|---|---|---|---|---|---|---|---|
| | val | test-A | test-B | dev | test | B@4 | CIDEr |
| BaseP | 85.8 | 88.8 | 81.7 | 83.4 | 83.5 | 37.3 | 125.5 |
| TEP-one-hot | 85.9 | 89.0 | 81.8 | 84.4 | 84.2 | 38.1 | 129.0 |
| TEP | 87.4 | 90.3 | 83.1 | 84.9 | 84.5 | 38.3 | 130.3 |

Table 11: TEP performance on VL-T5 backbone.

| Model | # Params | VQA Acc | RefCOCOg Acc | COCO Caption CIDEr |
|---|---|---|---|---|
| VL-T5-Single | 3P | 67.9 | 71.3 | **116.1** |
| VL-T5-All | P | 67.2 | 69.4 | 110.8 |
| VL-T5-All-TEP | P | **69.2** | **73.6** | 114.1 |

have low baseline performance on the detection task. Therefore, we have decided not to present the object detection task as our primary results, in line with the OFA (Wang et al., 2022) paper. We plan to extend the Musketeers to other backbones with stronger detection baselines in the future.

Table 12: Results on object detection for OFA and Musketeers in "base" size. "Multitask tuning is ✗" denotes single-task-tuned models.

| Model Type | Multitask tuning | # Samples | mAP | mAR |
|---|---|---|---|---|
| OFA | ✗ | 50k | 25.2 | 26.0 |
| Musketeers (BaseP) | ✓ | 50k | 24.9 | 25.9 |
| Musketeers (TEP) | ✓ | 50k | **25.8** | **26.4** |
| OFA | ✗ | 121k | 30.2 | 29.3 |
| Musketeers (BaseP) | ✓ | 121k | 29.8 | 28.7 |
| Musketeers (TEP) | ✓ | 121k | **30.4** | **29.4** |

**Six task training without object detection.** Although OFA-based models have low baseline performance for object detection, Table 13 demonstrates that incorporating object detection into joint training yields improvements in visual grounding performance. Based on these findings, we have decided to integrate object detection into the Musketeers' training scheme to enhance visual grounding results.

Table 13: Musketeers performance when removing object detection. We report results of Musketeers$_{base}$ trained on seven of six tasks(w/o object detection) in Subset$_{small}$. Adding object detection is beneficial to visual grounding performance.

| Prompt Type | # Detection Task | Visual Grounding | | | Visual Entailment | | Caption | | VQA |
|---|---|---|---|---|---|---|---|---|---|
| | | val | test-A | test-B | dev | test | B@4 | CIDEr | |
| TEP | ✗ | 87.4 | 89.7 | 82.8 | 84.8 | 84.5 | **38.5** | 129.3 | 70.3 |
| TEP | ✓ | **87.5** | **90.3** | **83.1** | **84.9** | 84.5 | 38.3 | **130.3** | **70.6** |

**Training & Inference Efficiency when usin TEP.** Table 14 demonstrates that Musketeer employs longer prompts as input, leading to approximately 10%-15% additional latency compared to the specialist OFA approach. However, Musketeer eliminates the need for multiple specialist models by leveraging fully-shared model parameters.

Table 14: Throughput (samples / second) comparison between OFA and Musketeer$_{Base}$ on one A100 GPU. Our analysis includes the average training and inference throughputs towards seven tasks (batch size 2 for each task). Musketeer demonstrates around 10%-15% extra latency overhead.

| Model | Training throughput | Inference throughput |
|---|---|---|
| OFA$_{Base}$ | 7.31 | 24.27 |
| Musketeer$_{Base}$ | 6.62 | 20.53 |

## A.3 IMPLEMENTATION DETAILS

**Training Details.** As suggested in (Yang et al., 2022; Wang et al., 2022; Li et al., 2022a), we initialize weights of Musketeer from pretrained model in (Wang et al., 2022). We directly evalu-

ate the joint-task trained model without any task-specific fine-tuning, in contrast to (Wang et al., 2022). To ensure consistency, all images are cropped to a size of $480 \times 480$, with $16 \times 16$ patches. We set the maximum text sequence length for both the encoder and decoder to 512, following (Wang et al., 2022). We use the AdamW optimizer (Loshchilov & Hutter, 2019) with $(\beta1, \beta2, \epsilon) = (0.9, 0.999, 1e-8)$ and a learning rate of $1e-4$, which is controlled by a linearly decayed scheduler with a warmup ratio of 0.01. Dropout regularization with a ratio of 0.1 and weight decay of 0.01 is also applied during training. Our models are trained with a batch size of 16 for each task (112 for seven tasks in total) on 8 A100 GPUs with 40GB memory. We update weights every 16 iterations. We further apply label smoothing with a factor of 0.1 and R-drop (Liang et al., 2021) to regularize the output distribution, preventing overfitting. We also use Automatic Mixed Precision (AMP) to train the model on FP16 precision, for faster computation.

**Ablation Details** Below are more details of our ablation study:

- **Evaluating the effect of task number on performance**: We explore the relationship between the number of tasks and multi-tasking performance by comparing the results of Musketeers trained on 3 (visual entailment, visual grounding, and caption), 5 (adding object detection and VQA), and 7 (further adding image classification and text summarization) tasks. Tab. 5 in the main text provides a detailed comparison of the results.

- **Replacing TEP subprompts with one-hot vector.** We replace subprompts except Instance Prompt in TEP with one-hot vectors. Tasks that share homogenous input/output formats will share the same one-hot vector for the corresponding TEP subprompts. For example, object detection and visual grounding will share the same one-hot vector for output formats. For other subprompts like data description, each task will hold an identical one-hot vector.

**Full TEP list** We provide full TEP lists for each task in Tab. 15. Subprompts for each task are specified by human, except Data description, which is obtained by ChatGPT.

Table 15: Full TEP lists for each task in structural form.

| Task | Subprompt | Content |
|---|---|---|
| Object detection | Data description | COCO, or the Common Objects in Context dataset, is a large-scale dataset for object detection, segmentation, and captioning. The dataset is commonly used to train and evaluate object detection algorithms. Annotating a dataset like COCO involves manually labeling the objects in each image with bounding boxes and class labels. This is typically done by trained annotators who use specialized software tools to draw the bounding boxes and assign the class labels to the objects in the images. |
| | Input format | A task prompt and an image containing target objects |
| | Output format | mutiple x0 + y0 + x1 + y1 |
| | Output description | mutiple bounding boxes (each consistsing of horizonal coordinates of leftupper points of target region + vertical coordinates of leftupper points of target region + horizonal coordinates of rightlower points of target region + vertical coordinates of rightlower points of target region ) |
| Image classfication | Data description | ImageNet is a large-scale dataset for image classification, object detection, and object segmentation. It contains over 14 million images, each labeled with the name of one of 1000 object categories. The images in ImageNet are annotated by human labelers, who have assigned a label to each image indicating the main object or concept depicted in it. The annotation process for ImageNet involves two steps: (1) determining the set of object categories to be used for labeling the images and (2) labeling the images with these categories. |
| | Input format | A Task prompt and an input image |
| | Output format | Text |
| | Output description | A class name this image describes |
| Visual grounding | Data description | RefCOCO is a dataset for referring expressions in images, which is built on top of the COCO dataset. Referring expressions are natural language phrases that refer to specific objects or regions in an image. For example, a referring expression might be "the dog in the center of the picture". Annotating a dataset like RefCOCO involves manually labeling the objects in each image with bounding boxes and class labels, as well as creating referring expressions that refer to specific objects or regions in the image. |
| | Input format | A Task Prompt, a text describe the target region and a image containing the target region |
| | Output format | x0 + y0 + x1 + y1 |
| | Output description | horizonal coordinates of leftupper points of target region + vertical coordinates of leftupper points of target region + horizonal coordinates of rightlower points of target region + vertical coordinates of rightlower points of target region |
| Image caption | Data description | In addition to object detection, the COCO dataset also includes annotations for image captioning. Image captioning involves generating a natural language description of the objects and scenes depicted in an image. To annotate a dataset for image captioning, annotators must assign a series of text descriptions to each image in the dataset. These descriptions should capture the key objects and scene elements present in the image, as well as their relationships and interactions. |
| | Input format | A Task Prompt and an input image |
| | Output format | Text |
| | Output description | Text that describe the input image |
| Visual entailment | Data description | SNLI-VE is a dataset for visual entailment, which is the task of determining whether a given natural language sentence is entailed by a given image. The SNLI-VE dataset is a large-scale dataset that includes over 200,000 images and more than 1.2 million sentence pairs. Annotating a dataset like SNLI-VE involves manually labeling the images with sentence pairs and labels indicating whether the sentences are entailed by the image.The sentences should be natural language sentences that are related to the content of the images, and the labels should indicate whether one sentence logically follows from the other given the information in the image. |
| | Input format | A Task Prompt, a condition text 1 , an implied result text 2 and an image |
| | Output format | Text |
| | Output description | Yes or no or maybe |
| VQA | Data description | VQAv2 is a dataset for visual question answering (VQA), which is a task that involves generating natural language answers to questions about images. The VQAv2 dataset is a large-scale dataset that includes over 200,000 images and more than 1.2 million questions and answers. Annotating a dataset like VQAv2 involves manually labeling the images with questions and answers. The questions should be natural language questions that are related to the content of the images, and the answers should be natural language responses that provide accurate and relevant information about the images. |
| | Input format | A Task Prompt , a question description text and an image |
| | Output format | Text |
| | Output description | Answers |
| Text summarization | Data description | Gigaword is a text corpus that is commonly used for training and evaluating text summarization models. The corpus consists of over a billion words of newswire text from various sources. To use Gigaword for text summarization, the text needs to be annotated with summary information. One common way to do this is by using the headline of each news article as a summary of the article itself. The headline is typically a short, one-sentence summary of the article's main point or topic, making it a natural choice for summarization. |
| | Input format | A Task Prompt and a Text |
| | Output format | Text |
| | Output description | Summary of input text |

