# OpenReview forum: "Musketeer: Joint Training/Inference for Multi-task Vision-Language Model with Task Explanation Prompts"
_ICLR.cc/2024/Conference — Submitted to ICLR 2024_

### Official Review · Reviewer_uHwb · 2023-11-01

**Soundness:** 3 good
**Presentation:** 2 fair
**Contribution:** 2 fair
**Rating:** 3
**Confidence:** 4

**Summary:**

The paper studies how to more effectively train a unified (sequence-to-sequence) vision-language model across multiple tasks. In prior work, each training example comes with a simple description of the intended task, e.g., “Which region does the text V describe”. The main argument in this work is that such simple descriptions may not be enough and more detailed and exhaustive task descriptions are beneficial.

For 7 pre-training tasks, the authors created detailed task descriptions consisting of data description, input/output format, and output description. Training with the detailed task descriptions shows improvement over training with simple and plain descriptions.

**Strengths:**

Training with more detailed task descriptions is a natural step to take for training instruction-following vision-language models. The authors verify that using such task descriptions indeed improves performance upon baselines such as OFA.

**Weaknesses:**

- Limited novelty and performance improvement

Replacing simple tasks descriptions with complex descriptions is a simple idea and has been successfully explored in language model literature [1]; thus the novelty is limited.

The paper shows that by using complex task descriptions, the model improves marginally upon the baseline on training tasks. It is not shown whether training on such task descriptions brings any new capacities (e.g., transfer to a new task with a new task description, or using task description to transfer to a new data domain, as is done in [1]); the core appeal of using complex task descriptions seems to be missing.

[1] Generalization via Declarative Instructions on 1600+ NLP Tasks. Wang et al. 2022



- Limited number of tasks and formats of task descriptions

The paper only studies 7 pre-training tasks, which makes the generalization of the conclusion questionable. For example, one big contributor in description is the inclusion of “data source”. Could it be because out of the 7 tasks, many data are from COCO so the model learns to utilize this information? What if the pre-training datasets all come from different image sources?

**Questions:**

- In Section 1.1, the paper states “For example, in visual grounding of some concept V , the prompt “Which region does the text V describe” requires the model to interpret “find” and represent the word “region” with sets of coordinates on the image plane, which do not have a meaningful (topologically consistent) representation in natural language.”  I am not quite sure what “interpret “find” and ‘represent the word “region”’ and “do not have a meaningful (topologically consistent) representation” means. Could the authors elaborate on this issue? The base prompts seem okay to me and are not ambiguous.

- How many "soft task vectors" does each task have for the learnable task vector baseline?

---

> ### Author Response · Authors · 2023-11-22
> **Response to Reviewer uHwb (Part 1/2)**
>
> We would like to express our sincere gratitude to Reviewer uHwb for your valuable time and effort in reviewing our work.  In our [General Response to All Reviewers](https://openreview.net/forum?id=jb37oaGZXy&nesting=2&sort=date-desc), we have thoroughly discussed the effectiveness of Task-Enhanced Pretraining (TEP) in unifying heterogeneous tasks, particularly in the context of few-shot transfer learning and zero-shot inference scenarios. In this response, we would like to delve deeper into addressing the additional weaknesses that have been identified.
>
>
> ## Q1: Motivation, novelty, and performance improvement issues.
> A1: Thanks for your comments. Here we hope to clarify several points that may be misunderstandings:
>
> ### 1. Joint training on multiple heterogeneous vision-language tasks while preserving single-task performance is quite challenging.
> Unlike NLP tasks which are homogeneous and suitable for large-scale concurrent training,  the computer vision tasks are generally heterogeneous, (e.g., object detection outputs a set of bounding boxes, which is fairly different from visual grounding where both a caption and the foreground object are described). Unified LLMs like GPTs have achieved a tremendous success due to the nature of sequence-to-sequence property in languages, yet a unified foundational generalist model (not with multiple heavily designed specific task heads) to perform a large variety of computer vision tasks such as object detection, image segmentation, image generation, 3D estimation, pose recognition, etcs. is not yet available.  Unifying heterogeneous tasks   in single shared models may result in  obvious performance drops (according to previous works such as Unified-IO and UniTAB) because of the task interference. When the number of heterogeneous tasks increases, the task interference will also become more significant. Therefore, single-task finetuning or task-specific adapters are necessary to achieve satisfying downstream task performance.  For example, the OFA model which is pre-trained on eight heterogeneous can not generate usable results in many pretraining tasks (e.g., pretrained OFA_base achieves ~70 CIDEr score on image caption, v.s. ~140 CIDEr for single task fine-tuned model ) without single task finetuning.
>
> ### 2. TEP is not a simple “complex” prompt.
> The main idea behind TEP is to use structural prompts to explicitly identify heterogeneous tasks in subprompt spaces.  In fact, as shown in Table 7 of the paper,  we have demonstrated that without structural form, even the more complex task description generated by ChatGPT with the richest information can not effectively mitigate the task interference problem.
>
>
> ### 3. The performance improvement of Musketeer and TEP is not marginal.
> We’d like to clarify that the performance improvements are not marginal from several perspectives:
> - First, as shown in Table 4 of the main text, Musketeer outperforms all the joint-trained models on heterogeneous tasks and even demonstrates competitive results to ~20 times larger models.
> - In addition, as shown in Table 2 and Table 3 of the main text, TEP-powered Musketeer outperforms BaseP consistently for all tasks and demonstrates comparable and even better results to single-task finetuned SOTA methods.  Please kindly note that fully eliminating the effects of task interference in heterogeneous tasks has proven to be challenging, as indicated by previous works such as Unified-IO and Pixel2SeqV2. However, to the best of our knowledge, Musketeer stands as the first endeavor to successfully address this challenge by unifying not only detection, visual grounding, and VQA tasks but also other tasks involving pure text or pure image inputs.
> - TEP’s improvements can be more substantial in few-shot and zero-shot scenarios.  As demonstrated in Appendix A.2 and General Response, TEP outperforms BaseP significantly in few-shot and zero-shot learning settings.

---

> > ### Author Response · Authors · 2023-11-22
> > **Response to Reviewer uHwb (Part 2/2)**
> >
> > ## Q2: What if the pre-training datasets all come from different image sources?
> > A2: Thanks. This might be a misunderstanding.   First, the main contribution of TEP is the structured form to provide explicit guidelines for heterogeneous tasks, not each subpromp.  For example,  as shown in Table 1 below, keeping the subprompt in a structured form while removing “data description” in TEP can already give a good performance boost, whereas using a complex prompt without structured subprompts performs poorly in our experiments (See Table 7 of the paper).
> >
> > **Table 1 Ablations on specific TEP subprompts. VG: Visual Grounding. VE: Visual Entailment. B@4: BLEU@4. Y/N: with / without data description.**
> > | **Prompt Type** | **Data Description** | **VG-val** | **VG-test-A** | **VG-test-B** | **VE-dev** | **VE-test** | **Caption-B@4** | **Caption-CIDEr** |
> > |-----------------|----------------------|-------------|---------------|---------------|------------|-------------|-----------------|:-----------------:|
> > | Base            | N                    | 85.8        | 88.8          | 81.7          | 83.4       | 83.5        | 37.3            | 125.5             |
> > | TEP             | N                    | 87.2        | 90.1          | 83.0          | 84.4       | 84.2        | 38.4            | 130.3             |
> > | TEP             | Y                    | 87.5        | 90.3          | 83.1          | 84.9       | 84.5        | 38.3            | 130.3
> >
> > Additionally, it is worth noting that our current experiments have already incorporated images from various sources. Specifically, our six vision-related tasks encompass images sourced from four distinct data sources. In our final version, we can further expand our experiments to include only one task from each data source, providing a more comprehensive analysis.
> >
> >
> >
> >
> > ## Q3. “The base prompts seem okay to me and are not ambiguous.”
> > A3: Thank you for the comment. This provides an opportunity to reiterate the main idea of our paper: Unlike TEP, the Base prompt lacks structural information that can be shared across various tasks.  This is reflected in the performance of the Base prompt trained model as the number of tasks increases. As shown in the table below (Table 2), as number of training task increasing (1->3->5->7), base prompt trained model's performance trend to decline, whereas the TEP-trained model consistently improves (In Table 5 of submitted paper), benefiting from the shared detailed task description information across tasks.
> >
> > **Table 2. Ablations on multi-task training with base prompts. VG: Visual Grounding. VE: Visual Entailment. B@4: BLEU@4.  #Task: number of training task, each task contains 50k samples**
> > | **Prompt Type** | **#Task** | **VG-val** | **VG-test-A** | **VG-test-B** | **VE-dev** | **VE-test** | **Caption-B@4** | **Caption-CIDEr** |
> > |-----------------|----------------------|-------------|---------------|---------------|------------|-------------|-----------------|:-----------------:|
> > | Base        	| 1                	| 88.5    	| 90.7      	| 83.3      	|  89.3  	| 89.2    	| 41.0        	| 138.2         	|
> > | Base        	| 3                	| 86.7    	| 89.7      	| 82.3      	|	84.4	| 84.2    	| 37.3        	| 126.0         	|
> > | Base        	| 5                	| 86.3    	| 89.3      	| 81.6      	|	84.2	| 84.1    	| 37.1        	| 125.7         	|
> > | Base        	| 7                	| 85.8    	| 88.8      	| 81.7      	| 83.4   	| 83.5    	| 37.3        	| 125.5         	|
> >
> >
> >
> > ## Q4. How many "soft task vectors" does each task have for the learnable task vector baseline
> > A4: We employ a  576-d soft task vector for each task. To investigate the potential impact of increasing the number of task vectors, we conducted experiments using 2 and 4 soft task vectors for each task. The results in Table 3 demonstrate that increasing the number of soft task vectors has a limited impact on Vision-Language multi-task learning.
> >
> > **Table 3. Ablations on number of soft task vectors. VG: Visual Grounding. VE: Visual Entailment. B@4: BLEU@4.**
> >
> > | **Prompt Type** |  **VG-val** | **VG-test-A** | **VG-test-B** | **VE-dev** | **VE-test** | **Caption-B@4** | **Caption-CIDEr** |
> > |-----------------|-------------|---------------|---------------|------------|-------------|-----------------|:-----------------:|
> > | 1 Soft vector     	| 84.9    	| 87.8      	| 80.7      	| 83.7   	| 83.6    	| 37.3        	| 125.5         	|
> > | 2 Soft vector     	|  85.1  	|  87.7    	|   80.9    	|   	83.6 |  	83.3   |	37.4     	|  	125.4    	|
> > | 4 Soft vector     	| 	84.8 	|	87.5   	|	80.1   	|	83.7	| 	83.5	| 	37.7    	|  	125.7

---

> > > ### Author Response · Authors · 2023-11-23
> > > **Last day reminder and looking forward to discussion**
> > >
> > > Dear reviewer uHwb,
> > >
> > > Thanks again for your valuable time and insightful comments. As the deadline for the Author/Reviewer discussion is approaching, it would be nice of you to let us know whether our answers have solved your concerns so that we can better improve our work. We are happy to provide any additional clarifications that you may need.
> > >
> > > Best regards!

---

### Official Review · Reviewer_nMNN · 2023-11-01

**Soundness:** 3 good
**Presentation:** 4 excellent
**Contribution:** 3 good
**Rating:** 8
**Confidence:** 4

**Summary:**

This paper investigates how to jointly finetuning a vision-language pretrained model onto several downstream tasks to achieve both optimal performance as well as task generalization. A model called Musketeer is proposed which utilizes Task Explanation Prompt (TEP) to reduce the interference among tasks, which helps the model to optimize each single-task better during multi-task downstream finetuning. The TEP contains sufficient task meta information, including data description, input/output format, output description and instance prompt. On downstream tasks, the Musketeer model achieves comparable or better single-task results over single-task finetuned baselines. Compared with multi-task finetuned baselines, Musketeer obtains much better results.

**Strengths:**

1. The research question is clear and important. Currently most pretrained-then-finetuned VL models still cannot achieve task generalization and SOTA performance together.
2. Detailed TEP of each downstream VL task are given, increasing the reproducibility of this work.
3. The baseline used is competitive, demonstrating the effectiveness of Musketeer model.
4. Abundant ablation analysis is conducted.

**Weaknesses:**

Since the TEP contains abundant downstream meta task information, if more discussion and experiment on zero-shot new task generalization, it will be much better.

**Questions:**

Since the OFA model not only unifies the VL tasks but also text-only tasks. Can Musketeer also be applied on text-only tasks? Is there any experimental evidence?

---

> ### Author Response · Authors · 2023-11-22
> **Response to Reviewer nMNN**
>
> Thank you  for your valuable and supportive reviews. We hope the below response can address your concerns.
>
>
> ## Q1:  If more discussion and experiment on zero-shot new task generalization, it will be much better.
> A1: We have provided an in-depth discussion and additional results of TEP and BaseP in both few-shot and zero-shot scenarios. Please see in [General Response to All Reviewers](https://openreview.net/forum?id=jb37oaGZXy&noteId=GV9Py9R2R3).
>
>
> ## Q2.  Can Musketeer also be applied on text-only tasks? Is there any experimental evidence?
> A2: Thanks for your comments. In fact, we have reported the Musketeer results on text-only task Gigawords in Table 3 of the main text. Additionally, in the zero-shot inference results in general response the Musketeer has also been demonstrated to be effective in other two unseen text-only datasets. We hope this can address your concern and We’ll add corresponding discussion in the future version.
>
>
> Should you have any further inquiries, please let us know and we are more than delighted to discuss with you and run more experiments for any pieces of your interests in our work.

---

> > ### Author Response · Authors · 2023-11-23
> > **Last day reminder and looking forward to discussion**
> >
> > Dear reviewer nMNN,
> >
> > Thanks again for your valuable time and insightful comments. As the deadline for the Author/Reviewer discussion is approaching, it would be nice of you to let us know whether our answers have solved your concerns so that we can better improve our work. We are happy to provide any additional clarifications that you may need.
> >
> > Best regards!

---

### Official Review · Reviewer_Dtxz · 2023-11-01

**Soundness:** 3 good
**Presentation:** 3 good
**Contribution:** 3 good
**Rating:** 6
**Confidence:** 4

**Summary:**

The paper proposes a new sequence-to-sequence vision-language model called Musketeer that can be trained jointly on multiple visual tasks using a shared set of parameters. The key idea is to use a novel Task Explanation Prompt (TEP) to reduce interference between tasks and allow the model to leverage shared structures. The TEP provides detailed natural language instructions about the dataset, input/output formats, output targets, etc. Experiments on 7 vision-language tasks like visual grounding, VQA, captioning etc show Musketeer matches or exceeds performance of task-specific models and other multi-task baselines. Without any task-specific tuning, Musketeer shows strong performance on all tasks using the descriptive power of TEPs to instantiate task-specific pathways at inference.

**Strengths:**

- The paper is well-written and clearly presented;
- The paper proposes a novel TEP approach to reduce multi-task interference using natural language specifications. It provides a unified architecture without any task-specific tuning or heads.
- It shows strong empirical results demonstrating effectiveness for diverse vision-language tasks comparing to baselines;
- Detailed experiments on the effects of each mixed dataset (vg, captain, ic, etc.) to the downstream have been provided across scales, which may benefit future researchers in the same area;

**Weaknesses:**

- TEP still relies on pretrained weights for initialization which can be expensive, the discussion regarding the additional cost might be good to provide;
- The hyper-parameter setting as well as the Needs carefully designed TEPs for new tasks which may require some expertise.
- The study on how well TEPs could transfer to unseen tasks is unknown;
- Some related works might also be good to include or discuss [1, 2, 3, 4];

[1] Dai, Wenliang et al. “InstructBLIP: Towards General-purpose Vision-Language Models with Instruction Tuning.” ArXiv abs/2305.06500 (2023);
[2] Shen, Sheng, et al. "Multitask vision-language prompt tuning." WACV 2024.
[3] Asai, Akari, et al. "Attempt: Parameter-efficient multi-task tuning via attentional mixtures of soft prompts." Proceedings of the 2022 Conference on Empirical Methods in Natural Language Processing. 2022.
[4] Liu, Haokun, et al. "Few-shot parameter-efficient fine-tuning is better and cheaper than in-context learning." Advances in Neural Information Processing Systems 35 (2022): 1950-1965.

**Questions:**

- Could the author explain more on the varied performance on VQA in table 4, will using full VQAv2 training data mitigate the problems?
- Could the author provide additional training cost including the pretrainining cost for the proposed methods in Table 3 and 4 for a comprehensive evaluations;

---

> ### Author Response · Authors · 2023-11-22
> **Response to Reviewer Dtxz (Part 1/2)**
>
> Thanks for your thoughtful review that will help us strengthen the manuscript. We have provided discussions of “how TEPs could transfer to unseen tasks” in [General Response to All Reviewers](https://openreview.net/forum?id=jb37oaGZXy&noteId=GV9Py9R2R3). Below we discuss other identified weaknesses.
>
> ## Q1: TEP still relies on pretrained weights for initialization which can be expensive, the discussion regarding the additional cost might be good to provide.
> A1: Thanks for your comments. We agree that the usage of pretrained weights may involve extra cost for pretraining. We’ll add corresponding discussion in our final version.  However, we hope to clarify that even without pretrained weights, TEP can still performs well on joint-training scenario for heterogeneous tasks.  In Table 1 we have provided results of Musketeer trained from scratch (without pretrained weights). As shown, Musketeer trained with TEP significantly outperforms BaseP and demonstrate usable performance even without pre-training. In addition, TEP is flexible and utilizes other pretrained weights (e.g.,  VL-T5 and OFA) for better overall performance.
>
> Table 1. Visual entailment performance of Musketeer trained with TEP and BaseP, both from scratch
> | **Prompt Type** | **dev** | **test**
> |----------------|---------|------------|
> | BaseP          | 56.7 |  56.9 |
> | TEP            | 73.1 | 73.0 |
>
>
> ## Q2: The hyper-parameter setting as well as the Needs carefully designed TEPs for new tasks which may require some expertise.
> A2:   Thanks. For the hyper-parameter settings, we hope to highlight that unlike many previous tasks which carefully design hyper-parameters and training metrics for each downstream task (e.g., OFA), we use fully-shared training settings (e.g., learning rates, epochs, and dropout ratio) for each task to simplify the hyper-parameter tuning efforts.  Besides, most hyper-parameter settings of TEP are following common practice of Transformer models of similar size (e.g., VL-T5).  Therefore, we hope to clarify that the  hyper-parameters settings of TEP are not carefully-designed and we believe its performance on each specific task can be further improved if we adopt extra careful hyper-parameter tuning.
> For the TEP design,  since the data description subprompt are generated by ChatGPT and verified by humans, we agree that it  may require some expertise. However, as shown in Table 2 below, the TEP without data description can already significantly outperform BaseP and the improvement by including data description is marginal, which means TEP can also be used without data description. Since other structural subprompts (instance prompt, I/O format and I/O description) are designed intuitively, we hope to point that TEP with no data description can also be used without much expertise.
>
>
>
>
>
>
>
>
>
> **Table 2. Ablations on specific TEP subprompts. VG: Visual Grounding. VE: Visual Entailment. B@4: BLEU@4. Y/N: with / without data description.**
> | **Prompt Type** | **Data Description** | **VG-val** | **VG-test-A** | **VG-test-B** | **VE-dev** | **VE-test** | **Caption-B@4** | **Caption-CIDEr** |
> |-----------------|----------------------|-------------|---------------|---------------|------------|-------------|-----------------|:-----------------:|
> | Base            | N                    | 85.8        | 88.8          | 81.7          | 83.4       | 83.5        | 37.3            | 125.5             |
> | TEP             | N                    | 87.2        | 90.1          | 83.0          | 84.4       | 84.2        | 38.4            | 130.3             |
> | TEP             | Y                    | 87.5        | 90.3          | 83.1          | 84.9       | 84.5        | 38.3            | 130.3
>
> ## Q3: Some related works might also be good to include or discuss.
> A3: Good advice. We’ll include corresponding discussions of  the mentioned works in our future version.

---

> ### Author Response · Authors · 2023-11-22
> **Response to Reviewer Dtxz (Part 2/2)**
>
> ## Q4: Can using full VQAv2 training data mitigate the problems in Table 4?
> A4: Thanks for advice. Due to the limitation on our computing resources, in order to complete the training within the specified rebuttal time window, we increase the VQAv2 training data utilization only from 44% to 60%. The outcomes reveal a 0.4% enhancement in performance on the VQAv2 test set. Table 3 further illustrates the correlation between increased VQAv2 data usage and improved model performance.  Based on these findings, we are confident that increasing the usage of VQAv2 data will mitigate the problem. Additionally, it's noteworthy that the Musketeer Large model, with only 472 M parameters, is significantly smaller than the parameter count of the Unified-IO_XLarge model, which stands at 2925 M.
>
> ## Q5: Discuss the additional training cost including the pretrainining cost for the proposed methods in Table 3 and 4.
> A5: Thanks for advice. We have provided pretraining costs comparison of Musketeer and other jointly-trained models without task-specific fine-tuning in the Table 3 below.  We’ll update the Table 3 and Table 4 considering pretraining cost and add more corresponding discussion in our final version.
>
>
> **Table 3 Pretraining costs comparison of Musketeer and other jointly-trained models without task-specific fine-tuning. Pretrain-LM: Dataset size of Pretrained language models. #Pretrain-others: numbers of samples for other pretrainining tasks.**
>
> | **Method**             | **#Params** | **Pretrain-LM** | **LM-source** | **#Pretrain-others** |
> |------------------------|-------------|-----------------|---------------|----------------------|
> | Flamingo               | -           | 182G            | M3W           | 339M                 |
> | UniTAB                 | -           | 160G            | ReBERTa       | 200K                 |
> | Unified-IO_Base        | 241M        | 750G            | C4            | 170M                 |
> | Unified-IO_Large       | 776M        | 750G            | C4            | 170M                 |
> | Unified-IO_XLarge      | 2925M       | 750G            | C4            | 170M                 |
> | Uni-Perceiver-v2_Base  | 308M        | 160G            | ReBERTa       | 30M                  |
> | Uni-Perceiver-v2_Large | 446M        | 160G            | ReBERTa       | 30M                  |
> | Musketeer_Base         | 182M        | 140G            | Pile          | 24M                  |
> | Musketeer_Large        | 472M        | 140G            | Pile          | 24M                  |

---

> > ### Author Response · Authors · 2023-11-23
> > **Last day reminder and looking forward to discussion**
> >
> > Dear reviewer Dtxz,
> >
> > Thanks again for your valuable time and insightful comments. As the deadline for the Author/Reviewer discussion is approaching, it would be nice of you to let us know whether our answers have solved your concerns so that we can better improve our work. We are happy to provide any additional clarifications that you may need.
> >
> > Best regards!

---

### Author Response · Authors · 2023-11-22
**General Response to All Reviewers**

# General Response to All Reviewers (Part 1/2)

We thank all the reviewers for their valuable feedback and great efforts, which have significantly contributed to the enhancement of our paper.  One notable inquiry regarding Musketeer revolves around its performance in scenarios with limited training data (few-shot) or even without any additional data (zero-shot), through the integration of data from other tasks. To address the reviewers' concerns about transferability of Musketeer and TEP to unseen tasks, we provide an in-depth discussion and additional results of TEP and BaseP (Base Prompt) in both few-shot and zero-shot scenarios.

## Few-shot concurrent finetuning results.
In order to investigate the performance of Musketeer with TEP in few-shot learning scenarios, we conducted training experiments with limited samples. Specifically, Musketeer_Base was trained on visual grounding (VG) and visual entailment (VE) using only 32 or 100 samples, while simultaneously fine-tuning the model with six other tasks that encompassed 50,000 samples each. Our findings, presented in Table 1 and Table 2, reveal that the TEP model surpasses the performance of BaseP pompted models on both the VE and VG tasks. For example, TEP outperforms BaseP by approximately 3% in the case of 32 VG samples and 4% in the case of 100 VG samples. These results indicate that TEP exhibits greater efficacy as a prompt for small data when utilized in a concurrent training configuration. For more detailed insights into our few-shot concurrent fine-tuning results, please refer to Appendix A.1.


**Table 1  Experiments of concurrent multi-task training for small data on visual grounding.**
| **Prompt Type** | **#Training Samples** | **val** | **test-A** | **test-B** |
|-----------------|-----------------------|---------|------------|------------|
| BaseP           | 32                    | 69.1    | 75.2       | 61.9       |
| TEP             | 32                    | 72.1    | 79.3       | 65.8       |
| BaseP           | 100                   | 75.1    | 80.3       | 70.5       |
| TEP             | 100                   | 79.2    | 84.5       | 74.1       |


**Table 2  Experiments of concurrent multi-task training for small data on visual entailment.**
| **Prompt Type** | **#Training Samples** | **dev** | **test**
|-----------------|-----------------------|---------|------------|
| BaseP          | 32                    | 67.2 |  67.8
| TEP             | 32                    | 69.4 | 69.5 |
| BaseP          | 100                   | 72.6 | 72.4 |
| TEP             | 100                   | 73.5 |73.8 |


## Zero-shot learning on unseen tasks.
In order to validate the enhanced transferability of Musketeer with TEP to unseen tasks, we conducted additional experiments focusing on the visual entailment (VE) performance. Specifically, we evaluated Musketeer models that were trained without any visual entailment samples. For this purpose, we trained the Musketeer_Base model with TEP and BaseP on six tasks from the subset_small dataset, excluding visual entailment. The results, as presented in Table 3, clearly demonstrate that the structured TEP prompts exhibit significant improvements in zero-shot performance, achieving a remarkable 10% higher accuracy compared to the BaseP models. This outcome suggests that TEP, when compared to BaseP (Base Prompt), effectively enhances the cross-task knowledge transfer within the joint-training scenario, thereby enabling superior zero-shot inference capabilities for previously unseen tasks.

**Table 3  Experiments of zero-shot learning on unseen visual entailment task.**
| **Prompt Type** | **#Training Samples** | **dev** | **test**
|-----------------|-----------------------|---------|------------|
| BaseP          | 0                   | 38.6  |  38.5 |
| TEP             | 0                   | 49.1 | 49.2 |

---

> ### Author Response · Authors · 2023-11-22
> **General Response to All Reviewers**
>
> # General Response to All Reviewers  (Part 2/2)
>
> ## Zero-shot inference on unseen datasets.
> We further demonstrate that TEP can also enhance the model zero-shot performance on unseen datasets of seen task. Specifically, we evaluate the seven-task trained Museketeer models  on unseen text summarization datasets tldr-news (https://huggingface.co/datasets/JulesBelveze/tldr_news) and news-summary (https://huggingface.co/datasets/argilla/news-summary), and the results are shown in Table 4. The structural TEP consistently demonstrates better zero-shot performance the BaseP on unseen datasets.
>
> **Table 4 Experiments of zero-shot learning on unseen datasets.**
>
> | **Dataset** | **Prompt Type** | **#Rouge-1** | **Rouge-2** | **Rouge-L** |
> |-------------|-----------------|--------------|-------------|-------------|
> | tldr-news   | BaseP           | 25.1         | 9.0         | 21.0        |
> | tldr-news   | TEP             | **29.1**     | **10.2**    | **27.4**    |
> | news-summary   | BaseP           | 33.3         | 14.1        | 30.8        |
> | news-summary   | TEP             | **40.2**     | **17.5**    | **36.9**    |
>
> ## Limitation of Musketeer on zero-shot open-task generation.
> Since Musketeer is trained on seven **heterogeneous vision/language tasks**, admittedly, Musketeer is not designed to aim at performing common open-task generalization to a form of totally unseen task such as image segmentation.
> However, according to above experiments, we hope to emphasize that TEP demonstrates significant improvements over BaseP in zero-shot inference on unseen tasks and datasets.
>
> In addition, we hope to highlight our main contribution, which lies in addressing the challenging issue of task interference during the joint-training of heterogeneous vision-language tasks. This is achieved through the utilization of structural TEP, accompanied by a balanced sampling and optimization scheme. In previous experiments, we have successfully demonstrated the effectiveness of structural TEP in mitigating task interference. Please also kindly note that Musketeer is also the first   joint-trained model  that can achieve comparable or even better performance on heterogeneous tasks including detection, grounding, image classification and other textual tasks.

---

### Meta-Review · Area_Chair_1w5o · 2023-12-04

**Metareview:**

This paper proposes to train a multitask vision-language model via appending more descriptive text descriptions of tasks. The main strength of this work is that the approach is sensible, simple, and leads to some empirical gains. The main drawback is that such approaches have been explored in the past, and the empirical improvements are mostly marginal.

**Justification For Why Not Higher Score:**

Marginal improvements over baseline; limited novelty.

**Justification For Why Not Lower Score:**

N/A

---

### Decision · Program_Chairs · 2024-01-16

Reject